# Sugar-Sweetened Beverage Consumption and Associated Factors in School-Going Adolescents of New Caledonia

**DOI:** 10.3390/nu11020452

**Published:** 2019-02-21

**Authors:** Guillaume Wattelez, Stéphane Frayon, Yolande Cavaloc, Sophie Cherrier, Yannick Lerrant, Olivier Galy

**Affiliations:** Interdisciplinary Laboratory for Research in Education, EA 7483, University of New Caledonia, Noumea BP R4 98851, New Caledonia; guillaume.wattelez@unc.nc (G.W.); stephanefrayon@hotmail.com (S.F.); yolande.cavaloc@unc.nc (Y.C.); sophie.cherrier@unc.nc (S.C.); yannick.lerrant@unc.nc (Y.L.)

**Keywords:** consumption behavior, knowledge, Melanesian, Pacific, physical activity, sugar-sweetened beverage, noncommunicable diseases, weight status, self-weight perception

## Abstract

This cross-sectional study assessed sugar-sweetened beverage (SSB) consumption and its associations with the sociodemographic and physical characteristics, behavior and knowledge of New Caledonian adolescents. The survey data of 447 adolescents from ages 11 to 16 years were collected in five secondary public schools of New Caledonia between July 2015 and April 2016. These data included measured height and weight, SSB consumption, sociodemographic characteristics, body weight perception, physical activity, and knowledge (sugar quantity/SSB unit; energy expenditure required to eliminate a unit) and opinions about the SSB‒weight gain relationship. Ninety percent of these adolescents declared regularly drinking SSBs. Quantities were associated with living environment (1.94 L·week^−1^ in urban environment vs. 4.49 L·week^−1^ in rural environment, *p* = 0.001), ethnic community (4.77 L·week^−1^ in Melanesians vs. 2.46 L·week^−1^ in Caucasians, *p* < 0.001) and knowledge about energy expenditure (6.22 L·week^−1^ in unknowledgeable adolescents vs. 4.26 L·week^−1^ in adolescents who underestimated, 3.73 L·week^−1^ in adolescents who overestimated, and 3.64 L·week^−1^ in adolescents who correctly responded on the energy expenditure required to eliminate an SSB unit, *p* = 0.033). To conclude, community-based health promotion strategies should (1) focus on the physical effort needed to negate SSB consumption rather than the nutritional energy from SSB units and (2) highlight how to achieve sustainable lifestyles and provide tools for greater understanding and positive action.

## 1. Introduction

Adolescent overweight and obesity are associated with lifestyle factors like low physical activity (PA) and frequent consumption of energy-dense foods high in saturated fats and refined carbohydrates [1]. Sugar-sweetened beverages (SSBs), in particular, are a frequent source of refined carbohydrates in adolescent nutrition [2,3,4]. According to the 2010 Dietary Guidelines for Americans, SSBs are “liquids that are sweetened with various forms of sugars that add calories. These beverages include, but are not limited to, soda and fruit drinks, and sports and energy drinks” [5]. SSBs are the largest source of added sugars and major energy contributors to diet [6]. Harnack et al. [2] showed that US school children drinking an average of 265 mL or more of soft drinks daily consumed almost 835 kJ more total energy per day than those drinking no soft drinks. Moreover, epidemiological studies have quantified the positive relationships between SSB consumption and long-term weight gain, type 2 diabetes mellitus and cardiovascular diseases [6,7].

SSB consumption is also related to weight status [8], self-weight perception and body dissatisfaction [8], and PA [9,10]. It may also be related to knowledge or opinions about SSB [6,11]. For example, Park et al. [6] found that SSB consumption was higher among adults who disagreed that SSBs contribute to weight gain, whereas knowledge about the energy content of regular soda was not associated with SSB intake. Conversely, other authors found that adolescents or young adults (college students) with the poorest nutrition knowledge had the highest SSB consumption [11].

The Pacific Island Countries and Territories (PICTs) are also facing problems of obesity, overweight and changing nutrition patterns [4]. Kessaram et al. [10] reported the high prevalence of overweight and obesity in adolescents, likely explained by environmental mutations and shifts in nutrition and PA [12]. Traditional diets of root crops, vegetables, fruits and fresh fish and meat have been steadily replaced by imported, processed, energy-dense, low-nutrient foods [13], including SSBs. In six PICTs, 42% of 13- to 15-year-old students declared drinking SSBs daily [10]. Pak et al. [14] assessed soft drink consumption and reported 84 L/person in Palau in 2011.

Obesity and comorbidities are a health concern for young adults in Oceania, especially those in New Caledonia [15]. New Caledonia is a French archipelago in the South Pacific located between 162° E–169° E longitude and 19° S–23° S latitude. Melanesians were the first inhabitants of this archipelago, and consecutive arrivals of different populations since the era of colonization have now provided a multi-ethnic society with several “ethnic communities” that are representative of the Pacific populations. Melanesian (39%), European (or Caucasian) (27%), Polynesian (8%), Asian (2.5%) and other populations live together [16]. In 1989, New Caledonia was divided into three provinces: the Southern Province, the Northern Province and the Loyalty Islands Province. The capital city Noumea (Southern Province) is nevertheless the center for most economic activities. As a consequence, Noumea and its suburbs are the only areas that can be considered urban. The lifestyles of urban and rural areas differ greatly, and cultural and socioeconomic differences are also observed among the main ethnic groups, especially in rural areas. Indeed, New Caledonia’s economy is overall on par with that of the Western world, especially in Noumea and other small towns, although half the population lives a traditional Pacific lifestyle. The traditional lifestyle is characterized by daily physical activities oriented toward agriculture, fishing or other activities to meet a family’s daily needs. Disparities are seen in all age groups including school-going adolescents. Whereas an “equitable school system” exists—i.e., offering every adolescent the same access to school and a standardized academic curriculum, as well as a health education program [17]—the differences between the ethnic communities may lead to dramatic consequences for health in future generations [18]. Although few studies have focused on 11- to 16-year-old adolescents in New Caledonia [13,19,20], the prevalence of overweight or obesity was found to be three times greater in New Caledonian adolescents than in French adolescents in the same age range [21]. In addition, Frayon et al. [13] showed that in this multi-ethnic society, the risk of being overweight/obese was significantly greater in Polynesian and Melanesian adolescents. Adolescent overweight was also associated with rural lifestyles and low socioeconomic status (SES) for girls and breakfast skipping for boys [22]. Melanesian adolescents were more physically active and had a higher body mass percentage, mainly in rural girls compared with urban girls [20]. PA and food intake, and adolescent knowledge about and consumption of SSBs remain inadequately explored in the New Caledonian population, but research suggests that socio-environmental variables should be taken into account. We therefore hypothesized that SSB consumption would be associated with ethnicity (i.e., being Melanesian), living environment (i.e., rural) and knowledge about SSBs (i.e., sugar contained in SSBs, energy expenditure to eliminate consumed SSBs and association with weight gain).

This study aimed to broaden the vision on health among the 11- to 16-year-old adolescents in New Caledonia by assessing their SSB consumption behaviors and the associations with individual and socio-environmental factors.

## 2. Materials and Methods

### 2.1. Data Collection and Participants

Our study was part of a community-based obesity prevention project conducted in five selected representative school sites in the three provinces (Northern Province, Southern Province and Loyalty Islands) of New Caledonia. The schools were selected on the basis of the following criteria: (1) a representative repartition of the schools between rural and urban areas (respectively, 63% and 37% of the population), and (2) sufficient school size (*n* > 200). Eight schools were eligible in the Southern Province (urban), four in the Northern Province (rural) and only one in the Loyalty Islands (rural). Selected schools were then randomly drawn among these eligible schools: two in the Southern Province, two in the Northern Province and one in the Loyalty Islands. The school and participant selection process are further described elsewhere [19].

Data were collected between July 2015 and April 2016 from 696 school-going adolescents (11–16 years old) during class time. In each school, only 95% of the expected participants responded due to absences or parental refusal. Adolescents with missing data (no height, no weight, no PA, inconsistent or no response: *n* = 221) were then excluded from this study, as were adolescents from an ethnic community other than Caucasian or Melanesian (mostly Asian and Polynesian: *n* = 28), as these subsamples were too small. The final total sample was 447 adolescents.

Parents gave informed written consent prior to the children’s participation in the study, in line with the legal requirements and the Declaration of Helsinki. The protocol was also approved by the Ethics Committee of the University of New Caledonia.

### 2.2. Measures

#### 2.2.1. Anthropometric Parameters

All anthropometric data, including body mass index (BMI) and the International Obesity Task Force (IOTF) criteria [23], were collected by trained staff (the senior researcher) in the school nurse’s office. Height was measured to the nearest 0.1 cm using a portable stadiometer (Leicester Tanita HR 001, Tanita Corporation, Tokyo, Japan). Weight was determined to the nearest 0.1 kg using a scale (Tanita HA 503, Tanita Corporation, Tokyo, Japan), with adolescents weighed in light clothing. Body mass index (BMI) was calculated by dividing weight in kilograms by height squared in meters.

The BMI standard deviation score (BMI-SDS) and percentile were calculated by the LMS method using the IOTF reference values. Weight status was defined according to the IOTF criteria, which are used to classify BMI values according to age and gender as thin (underweight), normal weight, overweight, or obese, based on adult BMI cutoffs at 18 years [23].

#### 2.2.2. Sociodemographic Characteristics

Ethnicity was self-reported by the adolescents using an anonymous survey tool and categorized as recommended in the report on New Caledonia [24] from the *Institut National de la Santé Et de la Recherche Médicale* (INSERM; National Institute of Health and Medical Research). Three SES categories were generated according to the National Statistics Socio-Economic Classification [25]: managerial and professional occupations (high), intermediate occupations (medium), and routine and manual occupations (low). According to the latest census in New Caledonia [26] and a European standard for the degree of urbanization [27], Noumea and its suburbs were classified as urban and the other areas were classified as rural.

#### 2.2.3. SSB Consumption

SSB consumption was assessed by first asking: “Do you drink SSBs?” Average amounts were then assessed with the question: “How many glasses (250 mL), cans (330 mL), small bottles (500 mL) or big bottles (1500 mL) do you drink per week on average?” The total SSB intake per week, converted to liters, was calculated by adding up the volumes of the total number of glasses, cans and bottles. Adolescents who answered “No” to the first question were considered as consuming 0 L per week.

#### 2.2.4. Self-Weight Perception

Adolescents were asked about their self-weight perception with the following question: “Given your age and height, would you say that you are about the right weight, too big, or too skinny?” [28]

#### 2.2.5. Physical Activity

The self-report International Physical Activity Questionnaire‒Short Form (IPAQ-SF) was filled out on the research day with the researchers’ assistance. The total number of days and minutes of PA was calculated for each participant [29]. Total PA time was converted to hours per week.

#### 2.2.6. Knowledge and Opinion about SSBs

Three parameters assessed participants’ knowledge about SSBs.

First, participants expressed their opinion about the effects of SSBs on weight by responding to this statement: “Drinking regular sodas, fruit drinks, sports or energy drinks, and other sugar-sweetened drinks can cause weight gain.” Responses were categorized as: agree (strongly/somewhat agree) or disagree (disagree completely, strongly/somewhat disagree).

Second, participants answered the following question to assess their “knowledge about sugar”: “How many sugar lumps does a regular can of, for example, non-diet cola contain?” Responses were categorized as: underestimated (<5 lumps), accurate (5–10 lumps), overestimated (>10 lumps), and don’t know.

Last, participants answered the following question to assess their “knowledge about energy expenditure”: “How much time do you need to run (at moderate speed) to eliminate the energy intake from drinking a regular can of, for example, non-diet cola?” The four response categories were: underestimated (<30 min), accurate (30–60 min), overestimated (>60 min), and don’t know.

### 2.3. Statistics

All analyses were conducted using R 3.1.0. [30] with a significance level of *p* = 0.05.

First, differences between categories for each explanatory factor were tested one by one. Differences were tested with the χ^2^ test or Fisher’s exact test (according to implementation conditions) for categorical variables and with means equality tests (Student, Welch or Wilcoxon) for numerical variables with binary factors. One-way ANOVA and the Kruskal-Wallis test were coupled with a post-hoc t-test and Nemenyi’s test, respectively, when needed for numerical variables with other factors. The PA‒quantity of SSB consumption correlation was determined and tested with Spearman’s test because Pearson’s test was not applicable.

The adolescents’ quantified SSB consumption was globally assessed by multiple regression and multifactorial ANOVA to detect associations, possible interactions and a confounding bias between factors. The significant pairwise interactions were included in the regression and ANOVA.

## 3. Results

Table 1 shows that socioeconomic factors were not discriminant for the “prevalence” of SSB consumption. Sex, living environment, ethnic community, SES and weight status were not significant factors and generally about 90% of the adolescents declared consuming SSBs. In addition, boys were significantly more physically active than girls (14.31 h·week^−1^ for boys vs. 12.18 h·week^−1^ for girls, *p* = 0.01), Melanesians were significantly more physically active than Caucasians (11.23 h·week^−1^ for Caucasians vs. 14.05 h·week^−1^ for Melanesians, *p* < 0.01), and adolescents were significantly more physically active in rural areas than in urban areas (13.64 h·week^−1^ in rural vs. 10.88 h·week^−1^ in urban, *p* < 0.01).

The analysis of factor interactions revealed significant pairwise interactions between SES and knowledge about energy expenditure (*p* < 0.01), weight status and knowledge about sugar (*p* = 0.01), weight status and knowledge about energy expenditure (*p* < 0.01), and PA and knowledge about energy expenditure (*p* = 0.02). Figure 1 shows the significant pairwise interactions between these categorical factors.

Table 2 shows that sex, SES, weight status, self-weight perception, knowledge about sugar, opinion about SSBs and weight gain, and PA were not significantly associated with quantity of SSB consumption. Conversely, living environment, ethnic community and knowledge about energy expenditure were significantly related to the quantity of SSBs consumed. In this general model, only interactions between SES and knowledge about energy expenditure (*p* = 0.01) and between PA and knowledge about energy expenditure (*p* = 0.02) were significant.

SSB consumption was significantly higher in rural environments than in urban environments, with respective values of 4.49 L·week^−1^ and 1.94 L·week^−1^ (*p* < 0.01, Table 2).

Melanesian adolescents consumed many more SSBs than their Caucasian counterparts, with respective values of 4.77 L·week^−1^ and 2.46 L·week^−1^ (*p* < 0.01, Table 2).

Weight status and self-weight perception were not significant factors of SSB consumption (*p* = 0.94 and 0.59, respectively). However, weight status significantly interacted with knowledge about sugar (Figure 1b, *p* = 0.01) and knowledge about energy expenditure (Figure 1c, *p* < 0.01), but these interactions were not satisfactory factors in the general ANOVA model (Table 2, *p* = 0.15 and *p* = 0.37, respectively).

The correlation between PA and the quantity of SSB consumption was ρ = 0.026 and this was not significant (*p* = 0.583). However, PA has to be taken into account to explain SSB consumption (even if *p* = 0.11) because of the significant interactions between PA and knowledge about energy expenditure (*p* = 0.02).

Knowledge about sugar was not a significant factor for SSB consumption (*p* = 0.12, Table 2), but most of the adolescents (192 for “accurate” and 153 for “overestimated”, i.e., 77%) reported knowing that there is a lot sugar in an SSB can. Conversely, knowledge about energy expenditure was a significant factor (*p* = 0.03), and the adolescents categorized as “don’t know” for energy expenditure knowledge consumed significantly more SSBs (6.22 L·week^−1^) than the others did (4.26 L·week^−1^ for adolescents who underestimated, 3.64 L·week^−1^ for adolescents who gave an accurate answer, and 3.73 L·week^−1^ for adolescents who overestimated the energy expenditure required to eliminate a SSB unit). The multiple regression in Table 2 shows that the higher the estimated expenditure, the lower the SSB consumption (B = −6.18 for “underestimated”, B = −7.75 for “accurate” and B = −8.29 for “overestimated”).

Opinion about the SSB consumption−weight gain relationship was not a significant factor for SSB consumption (B = 0.57, *p* = 0.20) and most of the adolescents (349, i.e., 78%) agreed with the statement “consuming SSBs can cause weight gain.”

## 4. Discussion

The main result of this study was that 90% of the New Caledonian adolescents claimed to regularly drink SSBs. Moreover, the entire population was concerned since they reported regular consumption regardless of sex, living environment, ethnic community, socioeconomic status and weight status. Notably, the quantity of SSBs was strongly correlated with social factors like living environment and ethnic community but weight status and self-weight perception were surprisingly not linked to quantity.

Knowledge about sugar was not significantly linked to quantity of SSB consumption but knowledge about energy expenditure was.

### 4.1. Sociodemographic Factors

In New Caledonia, the living environment, ethnic community and SES are interconnected factors. For example, most of the people living in rural environments have tribal lifestyles and are Melanesian, and most of the people from other ethnic communities (such as Caucasians) live in the urban environments of Noumea and its suburbs. Moreover, rural inhabitants are usually classified as low SES since they generally live of food crops [4]. Therefore, it is unsurprising that the quantity of consumed SSBs presented in Table 2 was significantly higher in rural environments, Melanesian adolescents, and low-SES adolescents as these factors are very much intertwined. This also may explain why, although the high-SES adolescents consumed significantly fewer SSBs than the low-SES adolescents, this factor was not significant in the general regression model (Table 2).

Adolescents living in rural areas reported significantly higher SSB consumption than those living in urban areas (4.49 L·week^−1^ vs. 1.94 L·week^−1^, respectively). This has also been observed in other countries. Indeed, Park et al. [6] reported that SSB intake differs significantly among US adults according to geographic region. They found that the proportion of adults who consumed SSBs ≥2 times per day, or more than 3.5 L·week^−1^, was highest among adults living in the East South Central region of the US [6]. Sharkey et al., who studied eating behaviors among rural and urban adults, found that the prevalence and high level of SSB consumption (≥3 cans or glasses of SSB per day, or more than 5 L·week^−1^) were greater among rural adults (52.4% prevalence and 17.7% high consumption) compared with urban counterparts (43.7% prevalence and 10.5% high consumption) [31]. The differences were partially explained by cultural norms, SSB availability, and/or state and local obesity prevention programs [6,31]. In New Caledonia, SSBs are widely available near all schools before, during and after the school day, and health education programs are delivered to all areas of the archipelago. Media access is nevertheless easier in the urban environments, and the media often provide strong warnings about the sugar in industrial products and its impact on health [10]. Moreover, Hughes and Lawrence [12] pointed out some of the factors that have changed Pacific Islanders’ behaviors, notably the acceptance and/or belief that foreign (imported) goods are superior. This may explain why Melanesian adolescents reported consuming significantly more SSBs than their Caucasian counterparts (4.77 L·week^−1^ and 2.46 L·week^−1^, respectively). In addition, a recent review evaluated the interventions targeting the consumption of high-sugar products by Pacific Island adolescents [10]. The authors reported that school and community-based interventions have had little success in reducing the consumption of these products for the moment [10].

As studies have shown, SSBs may also replace water as the daily beverage [32], and access to potable water might explain our observation in parts of rural New Caledonia [33]. In particularly remote areas, the water comes from desalination plants and/or the treated water may be of poor bacteriological quality [33]. People living in environments where access to drinking water is a problem might become suspicious of this essential natural resource, especially as the taste can be unpleasant. This might explain the preference for SSBs among some rural populations.

Several factors and their mutual interactions may in fact explain the prevalence of SSB consumption in rural environments and should be considered in the future: (1) drinking tap water may be not safe, so it is better to drink something else; (2) drinking SSBs or giving money to children to buy SSBs may be a way to ostensibly show personal earning power (more than buying bottled water); and (3) adolescent eating habits are influenced by media campaigns that communicate health and nutrition warnings, especially in urban environments.

### 4.2. Weight Status and Self-Weight Perception

A previous study reported that weight status might be linked to SSB intake [10], although another study found no association [34]. Researchers have explained this non-significant finding as underreported intake [35] or reduced intake by overweight and obese adolescents wanting to lose weight [34]. Our study shows that neither weight status (*p* = 0.94) nor self-weight perception (*p* = 0.59) was a significant explanatory factor for the quantity of SSB consumption (Table 2).

Previous studies conducted in New Caledonia have shown that people are aware that obesity can exacerbate noncommunicable diseases but they may not know the actual definition of obesity [36] and thus whether they are concerned. Adolescents have also shown a mismatch between weight perception and reality [19]. It is therefore possible that some of the overweight adolescents in the current study were not aware of their own weight status. In addition, overweight is overall associated with unhealthy lifestyles—not only SSB consumption but also other energy-dense food consumption. The total energy intake was unfortunately not assessed in this study and this might explain why weight status was not a significant explanatory factor for quantity of SSB consumption.

However, weight status related to knowledge may be linked to SSB intake (Figure 1) although, as shown in Table 2, the interactions between weight status and knowledge were not significant in the multiple regression (*p* = 0.15 for interactions between weight status and knowledge about sugar; *p* = 0.37 for interactions between weight status and knowledge about energy expenditure). Jasti et al. [11] found that overweight status modified the effect of SSB knowledge on SSB consumption. In their study, less SSB knowledge was associated with higher SSB consumption (OR = 3.56) only among overweight students, with no association in non-overweight students [11]. Our analysis of pairwise interactions (Figure 1b,c) showed that among the adolescents categorized as “don’t know” for sugar knowledge, those who were overweight had higher SSB intake than the others. Their overweight status may have been due in part to their lack of knowledge: they did not know how to behave healthily. On the contrary, the obese adolescents categorized as “don’t know” consumed fewer SSBs. This might be explained by medical monitoring: these adolescents had undoubtedly been warned against excessive SSB consumption and thus they refrained from drinking “too much” even if they had little knowledge about SSBs (sugar content and energy expenditure). In addition, the more highly the overweight adolescents estimated the sugar quantity in a can, the higher their SSB intake was. This may be due to fatalistic feelings or enjoyment being prioritized over health, especially when friends also drink SSBs: “There’s a lot of sugar in sweetened beverages, it’s unhealthy and it may cause weight gain, but sweetened beverages are so good and my friends are drinking a can… So, yes! I drink a can too!” [37]. But these observations should be carefully considered because of the small sample sizes in some subcategories.

### 4.3. Physical Activity

Bibiloni et al. [9] found that even though beverage intake and beverage total energy intake were positively associated with PA in Spanish adolescents, there was no association between PA and SSB intake (fruit drinks, soda or energy/sport beverages). In this study, inactive boys claimed that they drank more than 9 L·week^−1^ of SSBs (370.8 mL fruit drinks, 441.6 mL soda and 495.0 mL energy/sport beverages per day) and active boys claimed drinking more than 8 L·week^−1^ (338.3 mL fruit drinks, 474.0 mL soda and 337.1 mL energy/sport beverages per day). Inactive girls claimed drinking more than 6.5 L·week^−1^ of SSBs (312.6 mL fruit drinks, 400.4 mL soda and 233.3 mL energy/sport beverages per day) and active girls also claimed drinking more than 6.5 L·week^−1^ (269.7 mL fruit drinks, 365.4 mL soda and 330.0 mL energy/sport beverages per day) [9]. Our method differed from that of Bibiloni et al. Indeed, PA was not categorized but was processed as a continuous variable; for this reason, any comparisons must be cautiously made. However, when the active and inactive adolescents in New Caledonia were mixed, they reported drinking fewer SSBs than the Spaniards (4.25 L·week^−1^ in boys and 3.81 L·week^−1^ in girls, Table 2). In contrast to Bibiloni et al.’s study, our findings showed a positive correlation between PA and quantity of SSB consumption (B = 0.29, *p* ≤ 0.01). To explain this result, we need to consider that being active at school in a year-round subtropical climate requires high levels of hydration [38] and adolescents may prefer drinking other drinks than tap water in some remote areas (see Section 4.1.). In addition, the adolescents were significantly more active when they were boys (14.31 h·week^−1^ in boys vs. 12.18 h·week^−1^ in girls, *p* = 0.01), Melanesian (14.05 h·week^−1^ for Melanesians vs. 11.23 h·week^−1^ for Caucasians, *p* < 0.01), and living in rural areas (13.64 h·week^−1^ for rural adolescents vs. 10.88 h·week^−1^ for urban adolescents, *p* < 0.01) (Table 1). These results are in accordance with Zongo et al.’s study [20] and they support the observation that the interactions between sociocultural factors and behavioral factors are mutual in New Caledonia.

### 4.4. Knowledge and Opinion

Interestingly, knowledge about sugar was not a satisfactory explanatory factor (*p* = 0.12). This finding lines up with Park et al.’s findings [6] but is opposed to those of Jasti et al. [11]. The non-explanatory nature of this factor seems to indicate that thinking SSBs contain a lot of sugar does not affect the New Caledonian adolescents’ consumption. Notably, many of them gave an accurate answer or overestimated the quantity of sugar, and consequently they “knew”—or certainly thought they knew – that there was a lot of sugar, but that knowledge was not always totally accurate. In addition, despite this often vague knowledge, they consumed SSBs. However, knowledge about energy expenditure seemed to encourage the young people to consume less [39]. Indeed, the adolescents who were categorized as “don’t know” claimed that they drank many more SSBs than the others (6.22 L·week^−1^ for the “don’t know” vs. <4.3 L·week^−1^ for the others). Moreover, more than 78% reported agreeing with the statement “consuming SSBs can cause weight gain” and the adolescents who agreed that drinking SSBs can cause weight gain drank less than the others (3.77 L·week^−1^ for “agree” vs. 4.82 L·week^−1^ for “disagree”, *p* = 0.03). Similarly, Park et al. found many more individuals (≥29%) with high SSB consumption (more than 3.5 L·week^−1^, ≥2 servings per day) among those who did not think that drinking SSBs contributes to weight gain, compared with those who agreed about the SSB‒weight gain relationship (18.3% consumed 3.5L·week^−1^, ≥2 times per day) [6]. Nevertheless, opinion about the SSB consumption–weight gain relationship was not a satisfactory predictive factor (B = 0.57, *p* = 0.20), i.e., statistically, the B coefficient can be assumed to be zero. This seems to suggest that thinking SSBs can cause weight does not influence the SSB consumption in the adolescents of New Caledonia.

Our findings suggest that the key factor is “the notion of effort” related to SSB intake rather than “accurately knowing.” Indeed, the adjusted analyses showed that the “don’t know” adolescents consumed many more SSBs than the others, and the adolescents who thought that high energy expenditure was needed to eliminate the content of a SSB unit—that is, the “overestimators”—claimed consuming fewer SSBs than those who “underestimated” or were “accurate” (Table 2). Having an accurate answer seemed to be not that important as long as the notion of the physical effort needed to negate SSB intake had been learned. Moreover, even though most of the New Caledonian adolescents thought consuming SSBs can cause weight gain, they nonetheless drank them. Further studies are now needed to explain this observation.

### 4.5. Limitations and Strengths of the Study

This cross-sectional study does not provide evidence of causal associations or trends in the long term. However, the data were directly collected in the schools of interest: anthropometric measures were obtained during medical examination, providing more reliable assessments [19], and the survey was filled out on research days in the presence of researchers.

Adolescents were asked about their SSB consumption, PA, and knowledge and opinions about SSBs. The questions seemed to be easily understood and produced straightforward answers [39]. Yet some adolescents still gave aberrant answers probably because of a misunderstanding about the weekly average and the container volumes, and these answers were removed from analysis. The significant results were nevertheless based on robust statistical tests and thus were not impacted. Moreover, some of the adolescents might have taken sips from the drinks of the regular consumers. We assume that they took these small quantities into account in their responses, even though such quantities were probably negligible compared to drinking from one’s own container.

This study examined self-reported data, as well: SSB consumption, ethnic community and PA. These measures are subjective, especially PA quantification, but the associated surveys are commonly used to obtain this type of data [29,40]. This limitation might explain the non-significance of PA in explaining the quantity of SSB consumption. Future studies will therefore need to use objective data to assess PA.

As noted, the total energy intake was not assessed in this study even though weight status was introduced as an explanatory factor. It was therefore not possible to adjust the multiple regression with the total energy intake.

This study aimed to determine associations between SSB consumption and sociodemographic characteristics, including ethnic community, in adolescents of New Caledonia. However, for statistical reasons—that is, the sample size—only the two main ethnic communities were retained, i.e., Caucasian and Melanesian. This may have introduced a bias for comparisons with the general population of New Caledonia.

## 5. Conclusions

This study provides insight into the links between SSB consumption and socioeconomic factors, actual weight status and self-weight perception, PA, and knowledge and opinions about SSBs in New Caledonian adolescents.

First, this study showed that SSB consumption in New Caledonia is strongly associated with ethnic community and living environment. This result should contribute to developing community-based health promotion strategies.

Second, New Caledonian adolescents seem to have an inadequate understanding of the associations between PA, food consumption and health, which raises questions about health communication campaigns from an educational point of view [4,6]. Notably, knowing how much physical effort is needed to negate an SSB seems to have more influence than knowing the sugar quantity in an SSB can. As noted, the “notion of effort” related to SSB intake may best drive adolescents to moderate SSB consumption.

Yet more studies are needed on how New Caledonian youths relate to issues of health, nutrition and physical activity. Longitudinal and qualitative studies should examine why adolescents drink so many SSBs even when they “know”—that “knowledge” often being quite inexact—that there is too much sugar in them and they think SSBs can cause weight gain.

## Figures and Tables

**Figure 1 nutrients-11-00452-f001:**
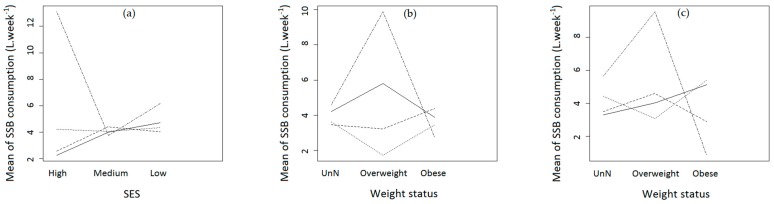
Pairwise factor interaction plots for SSB consumption: (**a**) Socioeconomic status (SES) vs. knowledge about energy expenditure, *p* = 0.005; (**b**) Weight status vs. knowledge about sugar, *p* = 0.011; (**c**) Weight status vs. knowledge about energy expenditure, *p* = 0.002. Type of line corresponds to knowledge. Solid line: adolescents who overestimated; dashed line: adolescents who gave accurate answers; dotted line: adolescents who underestimated; dotted-dashed line: adolescents who did not know.

**Table 1 nutrients-11-00452-t001:** Population: proportions of weight status, weight perception, SSB consumption (yes or no) and physical activity according to sex, living environment, ethnic community, socioeconomic status (SES) and weight status (*n* = 447).

			Weight Status (%)				Self-Weight Perception (%)				SSB Consumption (%)			Physical Activity (h·week^−1^)	
		Size	UnN *	Overweight	Obese	*p*-Value ^#^	Too Skinny	Normal	Too Big	*p*-Value ^#^	Yes	No	*p*-Value ^#^	Mean ± SD	*p*-Value ^†^
**Sex**	**Boys**	194	65.5	19.1	15.5	0.453	13.9	67.5	18.6	0.393	90.7	9.3	0.946	14.31 ± 9.31	0.011
	**Girls**	253	64.0	23.3	12.7		16.6	61.3	22.1		90.9	9.1		12.18 ± 7.81	
**Living environment**	**Rural**	361	60.9	23.0	16.1	0.002	15.2	63.7	21.1	0.873	91.7	8.3	0.278	13.64 ± 8.68	0.007
	**Urban**	86	80.2	15.1	4.7		16.3	65.1	18.6		87.2	12.8		10.88 ± 7.62	
**Ethnic community**	**Caucasian**	149	75.8	12.8	11.4	0.001	15.4	67.8	16.8	0.354	87.9	12.1	0.183	11.23 ± 8.67	0.001
	**Melanesian**	298	59.1	25.8	15.1		15.4	62.1	22.5		92.3	7.7		14.05 ± 8.34	
**SES**	**Low**	205	58.1	24.9	17.1	0.041	16.1	62.9	21.0	0.831	89.8	10.2	0.751	12.69 ± 8.03	0.586
	**Medium**	115	65.2	20.9	13.9		17.4	61.7	20.9		92.2	7.8		13.21 ± 8.94	
	**High**	127	74.8	16.5	8.7		12.6	67.7	19.7		91.3	8.7		13.68 ± 9.28	
**Weight status**	**UnN ***	289					23.5	71.6	4.8	<0.001	92.4	7.6	0.289	13.25 ± 8.62	0.731
	**Overweight**	96					1.0	56.3	42.7		88.5	11.5		13.18 ± 8.36	
	**Obese**	62					0.0	40.3	59.7		87.1	12.9		12.32 ± 8.56	

* UnN: underweight and normal. **^#^** 𝛘^2^ test or Fisher’s exact test. ^†^ Binary factors: Student’s or Welch’s *t*-test. Other factors: one-way ANOVA.

**Table 2 nutrients-11-00452-t002:** SSB consumption: quantity, multifactorial regression and ANOVA (*n* = 447).

			Quantity (L·week^−1^)		Multiple Regression ^1^ (R^2^ = 0.25; R^2^_adj_ = 0.18)		Multifactorial ANOVA ^1^
		Size	Mean ± SD	*p*-Value *	B ± SE (95% CI)	*p*-Value	*p*-Value
**Sex**	**Boys**	194	4.25 ± 4.30	0.258	0.50 ± 0.38 (−0.24;1.24)	0.187	0.187
	**Girls**	253	3.81 ± 4.00				
**Living environment**	**Rural**	361	4.49 ± 4.31	<0.001			0.001
	**Urban**	86	1.94 ± 2.39		−1.67 ± 0.51 (−2.67;−0.66)	0.001	
**Ethnic community**	**Caucasian**	149	2.46 ± 3.23	<0.001			<0.001
	**Melanesian**	298	4.77 ± 4.32		1.91 ± 0.43 (1.07;2.75)	<0.001	
**SES** ^†^	**Low**	205	4.44 ± 4.17 ^a^	0.014	−11.08 ± 3.23 (−17.42;−4.74)	0.001	0.607
	**Medium**	115	4.20 ± 3.83 ^a,b^		−11.06 ± 3.19 (−17.34;−4.79)	0.001	
	**High**	127	3.11 ± 4.23 ^b^				
**Weight status**	**Underweight and normal**	289	3.86 ± 3.91	0.392			0.943
	**Overweight**	96	4.51 ± 4.75		1.86 ± 2.62 (−3.28;7.01)	0.477	
	**Obese**	62	3.87 ± 4.09		−1.54 ± 3.09 (−7.61;4.53)	0.618	
**Weight perception**	**Too skinny**	69	4.31 ± 4.30	0.748			0.590
	**Normal**	286	3.99 ± 4.09		−0.35 ± 0.53 (−1.39;0.69)	0.511	
	**Too big**	92	3.81 ± 4.18		−0.75 ± 0.73 (−2.19;0.69)	0.305	
**How many sugar lumps in a SSB can?**	**Do not know**	24	5.23 ± 5.72	0.071			0.124
	**Underestimate**	78	3.25 ± 3.61		−0.52 ± 1.30 (−3.08;2.03)	0.689	
	**Accurate**	153	3.53 ± 3.59		−0.30 ± 1.26 (−2.77;2.16)	0.808	
	**Overestimate**	192	4.53 ± 4.42		−0.11 ± 1.23 (−2.53;2.32)	0.932	
**How much running is needed to eliminate the sugar contained in a SSB can?**	**Do not know**	27	6.22 ± 6.39	0.174			0.033
	**Underestimate**	137	4.26 ± 4.23		−6.18 ± 3.09 (−12.26;−0.10)	0.046	
	**Accurate**	201	3.64 ± 3.75		−7.75 ± 3.03 (−13.71;−1.79)	0.011	
	**Overestimate**	82	3.73 ± 3.74		−8.29 ± 3.15 (−14.47;−2.10)	0.009	
**Consuming SSBs can cause weight gain**	**Agree**	349	3.77 ± 4.06	0.027			0.204
	**Disagree**	98	4.82 ± 4.28		0.57 ± 0.45 (−0.31;1.46)	0.204	
**Physical activity**		447			0.29 ± 0.11 (0.08;0.50)	0.006	0.111

^1^ Analyses adjusted by the significant pairwise interactions: SES and knowledge about energy expenditure (*p* = 0.011), weight status and knowledge about sugar (*p* = 0.147), weight status and knowledge about energy expenditure (*p* = 0.367), and physical activity and knowledge about energy expenditure (*p* = 0.017). * Binary factors: Student’s, Welch’s or Wilcoxon’s test. Other factors: one-way ANOVA or Kruskal-Wallis’ test. ^†^ One or several subscripted letters in a cell indicate a post-hoc pairwise t-test result. Similar letters: no significant difference between two groups. Post-hoc pairwise *t*-test *p*-values: *p* = 0.604 for low and medium, *p* = 0.013 for low and high and *p* = 0.081 for medium and high.

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
