# Peer review of "Sugar-Sweetened Beverage Consumption and Associated Factors in School-Going Adolescents of New Caledonia"

_nutrients, 2019, doi:10.3390/nu11020452_

Reviewer 1 Report

This is a well-written paper but I have two general recommendations for improvement.  

First, while you point out some comparisons to other populations, it would be helpful to report the data (e.g., percentage use in various populations) along with mentioning its comparable.  I would also like to see more comaparisons to Eastern, European, and Western (USA) data where available so the reader can see how your data compare.  You could even include a table to this effect which would be helpful. Example, you mention studies from the US, Spain, etc. and describe the general findings, but data is needed for comparison where possible.

I would also suggest reviewing the journal format. There shouldn't be a large amount of empty space in the paper as shown on page 7, even when inserting tables.

Author Response

Responses to reviewer 1

This is a well-written paper but I have two general recommendations for improvement. 

We thank the reviewer for his/her appraisal and we have done our best to address the concerns. We hope this current version will convince the reviewer that the manuscript is now suitable for publication.

Q1a. First, while you point out some comparisons to other populations, it would be helpful to report the data (e.g., percentage use in various populations) along with mentioning its comparable. I would also like to see more comaparisons to Eastern, European, and Western (USA) data where available so the reader can see how your data compare. 

R1a. Thank you for this proposal. We have added data to the manuscript when available and therefore more comparisons are presented in the text. We have attempted to do this in the clearest way possible so that readers can easily compare our results with those of other studies. For this reason, we have converted the results of the other studies into the units we use: mL per day, servings per day, etc. were converted to L per week. Please see:

Line 235 (4.1) : “They found that the proportion of adults who consumed SSBs ≥ 2 times per day, or more than 3.5 L.week-1, was highest among adults living in the East South Central region of the US [4]. Sharkey et al., who studied eating behaviors among rural and urban adults found that the prevalence and high level of SSB consumption (≥ 3 cans or glasses of SSB per day, or more than 5 L.week-1) were greater among rural adults (52.4% prevalence and 17.7% high consumption) compared with urban counterparts (43.7% prevalence and 10.5% high consumption) [30].”

Line 275 (4.2) : “In their study, less SSB knowledge was associated with higher SSB consumption (OR=3.56) only among overweight students, with no association in non-overweight students [9].”

Line 297 (4.3) : “Bibiloni et al. [7] found that even though beverage intake and beverage total energy intake were positively associated with PA in Spanish adolescents, there was no association between PA and SSB intake (fruit drinks, soda or energy/sport beverages). In this study, inactive boys claimed that they drank more than 9 L.week-1 of SSB (370.8 mL fruit drinks, 441.6 mL soda and 495.0 mL energy/sport beverages per day), whereas active boys claimed drinking more than 8 L.week-1 (338.3 mL fruit drinks, 474.0 mL soda and 337.1 mL energy/sport beverages per day). Inactive girls claimed drinking more than 6.5 L.week-1 of SSB (312.6 mL fruit drinks, 400.4 mL soda and 233.3 mL energy/sport beverages per day) and active girls claimed drinking more than 6.5 L.week-1 too (269.7 mL fruit drinks, 365.4 mL soda and 330.0 mL energy/sport beverages per day) [7]. In contrast, our findings showed a positive correlation between PA and quantity of SSB consumption (, P≤0.01). Indeed, being active at school in a year-round subtropical climate requires high levels of hydration [36] and could explain this positive correlation. In addition, the adolescents were significantly more active when they were boys, Melanesian and living in rural areas (Table 1). These results are in accordance with Zongo et al.’s study [19] and they support the observation that the interactions between sociocultural factors and behavioral factors are mutual in New Caledonia.”

Line 314 (4.4) : “Similarly, Park et al. found many more individuals (≥29%) with high SSB consumption (more than 3.5 L.week-1, ≥2 servings per day) among those who did not think that drinking SSBs contributes to weight gain, compared with those who agreed about the SSB‒weight gain relationship (18.3% consumed 3.5L.week-1, ≥2 times per day) [4].”

Q1b. You could even include a table to this effect which would be helpful. Example, you mention studies from the US, Spain, etc. and describe the general findings, but data is needed for comparison where possible.

R1b. Adding a table to easily compare our study to others is a good idea. However, the methods, units, categories and aims of the various studies were very different (i.e. mL per day, servings per day, etc.). Despite the interesting information such a table would provide, it would require lengthy explanations in methods (how to homogenize the units, what about the categories? How to convert categories in continuous data and vice-versa, etc.), and this might not be of particular interest to readers. Therefore, in the present manuscript, we have very carefully detailed the numbers in L per week in the text. Please see the answer to R1a.

Q2. I would also suggest reviewing the journal format. There shouldn't be a large amount of empty space in the paper as shown on page 7, even when inserting tables. 

R2. As requested by the reviewer, we have carefully re-read the author recommendations and worked on the manuscript. We hope that the paper is acceptable for Nutrients journal in its current form.

Reviewer 2 Report

This paper investigated the sugar-sweetened beverage consumption and associated factors in school-going adolescents of New Caledonia. I should point out several limitations. 

Several points should be addressed:

1. The introduction is very weak. Why do we need to know about New Caledonia? What is the connection between the introduction and the proposed hypothesis? The introduction itself does not provide sufficient information to back up the research hypothesis. Thus, the connection is very poor. 

2. Frankly speaking, I really do not think you need figure 1. I do not think conducting pairwise factor interaction plots would provide useful information to the readers. 

3. What about those who shared their SSB with friends? Did you classify those individuals who might just take a sip with those who were regular consumers? 

4. If you report beta coefficient as part of your statistical analyses, you should report standard errors (SE) as well. 

5. Several sentences included some grammatical mistakes. Please check.

6. I must say that there are many papers already investigating the associated factors with SSB worldwide. The authors did not make a strong case to convince the readers/reviewers that why we want to know about New Caledonia. Besides, this is a cross-sectional study. I must say that the the cross-sectional design does not provide much information to the readers to understand the trends in the long term. 

Author Response

Responses to reviewer 2

This paper investigated the sugar-sweetened beverage consumption and associated factors in school-going adolescents of New Caledonia. I should point out several limitations.

We thank the reviewer for his/her time and effort in assessing this paper with these relevant comments. We have attempted to address all the points he/she has raised. We hope this current version will convince the reviewer that the manuscript is acceptable for publication in Nutrients.

Several points should be addressed:

Q1. The introduction is very weak. Why do we need to know about New Caledonia? What is the connection between the introduction and the proposed hypothesis? The introduction itself does not provide sufficient information to back up the research hypothesis. Thus, the connection is very poor.

R1. To provide readers with a better understanding of our research, we have worked to improve our introduction and now include information on the context of New Caledonia; please see lines 56. In addition, we have added a hypothesis in line with the context and the introduction has been reworded; please see lines 84. We sincerely hope that the connection between the research topic, the New Caledonia context, and our working hypothesis is now clearer for the reader.

Q2. Frankly speaking, I really do not think you need figure 1. I do not think conducting pairwise factor interaction plots would provide useful information to the readers. 

R2. Factor interaction plots are needed in multiple regressions in order to conduct a comprehensive statistical analysis. This enables understanding of the ins and outs of the issue at hand. In our study, some pairwise factor interactions were significant, which is why they were included as factors in the multiple regression (Table 2). Moreover, this kind of analysis gave us a few clues for the topic under study. Knowing that there are interactions between factors is interesting but seeing and understanding how the factors interact enables measuring the scale of the interactions for each category. This then helps clarify and explain the behavior of certain individuals (4.2 line 281). We included Figure 1 for this reason: We thought it would help the readers to have a better understanding of our hypotheses and discussion.

Q3. What about those who shared their SSB with friends? Did you classify those individuals who might just take a sip with those who were regular consumers? 

R3. Unfortunately, we did not do that. However, each adolescent in our study answered the following question: “How many glasses (250 mL), cans (330 mL), small bottles (500 mL) or big bottles (1,500 mL) do you drink per week on average?” We assume that they took into consideration all quantities drunk in their report. In any case, it would have been impossible to identify how much they drank when sharing a sip. Nevertheless, we have added a sentence in the limitations to highlight this issue.

Please see lines 332: “Moreover, some adolescents might take a sip with those who are regular consumers. We assume that they took in account this quantity in their answers even if this quantity is probably negligible compared to their consumption in their own containers.”

Q4. If you report beta coefficient as part of your statistical analyses, you should report standard errors (SE) as well. 

R4. Thank you for this point. Even though confidence intervals (CI) are retrieved from standard errors (SE), they have been added to Table 2 (B ± SE (95% CI)).

Q5. Several sentences included some grammatical mistakes. Please check.

R5. We have carefully checked the manuscript with a professional English editing service of the university and we sincerely hope the revised version is now acceptable for Nutrients journal.

Q6. I must say that there are many papers already investigating the associated factors with SSB worldwide. The authors did not make a strong case to convince the readers/reviewers that why we want to know about New Caledonia. Besides, this is a cross-sectional study. I must say that the the cross-sectional design does not provide much information to the readers to understand the trends in the long term. 

R6. The culture of New Caledonia is very close to the French culture and the two school systems are very similar. Nevertheless, the population of New Caledonia is very different from that of metropolitan France. In addition, overweight and obesity are proportionally three times greater among New Caledonia adolescents than among French adolescents of the same age range. Overweight appears at a younger age at a striking scale, and this results in high costs for the healthcare system. If this continues, what will we find in 20 years? To date, there have been few research studies about nutrition and lifestyle behaviors in the Pacific and especially in New Caledonia. One of the priorities of our lab is therefore to work on lifestyle issues among Melanesian youth. We now explain this more fully in the introduction; please see lines 65-84.

Round  2

Reviewer 2 Report

Thanks for the authors' effort to improve the quality of this paper. The newly revised manuscript has addressed all concerns and questions. It should be ready for publication. 

Author Response

Many thanks for your time spend to do this expertise